# GraphTOP: Graph Topology-Oriented Prompting for Graph Neural Networks

**Xingbo Fu**
University of Virginia
Charlottesville, VA, USA
xf3av@virginia.edu

**Zhenyu Lei**
University of Virginia
Charlottesville, VA, USA
vjd5zr@virginia.edu

**Zihan Chen**
University of Virginia
Charlottesville, VA, USA
brf3rx@virginia.edu

**Binchi Zhang**
University of Virginia
Charlottesville, VA, USA
epb6gw@virginia.edu

**Chuxu Zhang**
University of Connecticut
Storrs, CT, USA
chuxu.zhang@uconn.edu

**Jundong Li**
University of Virginia
Charlottesville, VA, USA
jundong@virginia.edu

## Abstract

Graph Neural Networks (GNNs) have revolutionized the field of graph learning by learning expressive graph representations from massive graph data. As a common pattern to train powerful GNNs, the "pre-training, adaptation" scheme first pre-trains GNNs over unlabeled graph data and subsequently adapts them to specific downstream tasks. In the adaptation phase, graph prompting is an effective strategy that modifies input graph data with learnable prompts while keeping pre-trained GNN models frozen. Typically, existing graph prompting studies mainly focus on *feature-oriented* methods that apply graph prompts to node features or hidden representations. However, these studies often achieve suboptimal performance, as they consistently overlook the potential of *topology-oriented* prompting, which adapts pre-trained GNNs by modifying the graph topology. In this study, we conduct a pioneering investigation of graph prompting in terms of graph topology. We propose the first **Graph T**opology-**O**riented **P**rompting (GraphTOP) framework to effectively adapt pre-trained GNN models for downstream tasks. More specifically, we reformulate topology-oriented prompting as an edge rewiring problem within multi-hop local subgraphs and relax it into the continuous probability space through reparameterization while ensuring tight relaxation and preserving graph sparsity. Extensive experiments on five graph datasets under four pre-training strategies demonstrate that our proposed GraphTOP outshines six baselines on multiple node classification datasets. Our code is available at https://github.com/xbfu/GraphTOP.

## 1 Introduction

Graphs are ubiquitous in a wide range of real-world scenarios, such as social networks [43, 61], knowledge graphs [41], traffic networks [32], and healthcare [3, 7]. To gain deep insights from tremendous graph data, numerous graph learning models have been developed in recent years. Among these efforts, Graph Neural Networks (GNNs) [1, 11, 23, 33, 38, 42, 46] are a prevalent tool for modeling graph data and have shown great prowess in different graph-related downstream tasks, including node classification [28, 56], link prediction [16, 62], and graph classification [25, 63]. Traditionally, GNN models are trained in a supervised manner. However, the supervised manner relies heavily on sufficient labeled graph data, which may be infeasible in the real world. Furthermore, the trained GNN models cannot be well generalized to other downstream tasks, even on the same graph data. These two critical issues hinder further deployments of GNN models in practice.

39th Conference on Neural Information Processing Systems (NeurIPS 2025).

To address the above issues, the "pre-training, adaptation" scheme has been widely adopted by a cornucopia of studies [5, 15, 27, 35, 36, 52, 57, 8]. Typically, they first train GNN models on pre-training tasks in an unsupervised manner, followed by adapting the pre-trained GNN models to specific downstream tasks. For instance, a GNN model can be pre-trained via link prediction and later adapted for node classification as the downstream task. During the adaptation phase, the goal is to bridge the objective gap between pre-training and downstream tasks. Inspired by recent prompt tuning approaches in natural language processing [21, 58] and computer vision [18, 48], graph prompting [9] has become an effective adaptation strategy to adapt pre-trained GNN models for downstream tasks by modifying the input graph data with trainable prompt vectors while keeping the pre-trained GNN models frozen. Generally, the existing graph prompting methods are *feature-oriented* — they design and learn graph prompts mainly by applying them to node features [5, 36] or hidden representations [27, 51, 52].

While the feature-oriented design is intuitive in graph prompting methods, they mostly overlook graph topology — another essential component in graphs that fundamentally distinguishes graph data from image data and text data — when designing graph prompts. Notably, graph representations are not only dependent on feature information but also determined by graph topology. Numerous efforts have demonstrated the significant impact of graph structures on graph-related tasks, particularly node classification [6, 20, 26, 37, 55]. Unfortunately, the potential of *topology-oriented* prompting for pre-trained GNN models remains unexplored in existing studies. Therefore, it is natural to pose the question: *How should we design a topology-oriented prompting framework to effectively adapt a pre-trained GNN model for downstream tasks?*

To answer this question, we conduct the pioneering investigation of graph prompting in terms of graph topology. We propose GraphTOP — the first **Graph Topology-Oriented Prompting** framework to adapt pre-trained GNN models for downstream tasks, particularly for node classification. In GraphTOP, we formulate topology-oriented prompting as an edge rewiring problem and relax it into the continuous probability space via reparameterization. To ensure computational feasibility in practice, we propose subgraph-constrained topology-oriented prompting by restricting edge rewiring between each target node and other nodes within its multi-hop subgraphs. In the end, we design the optimization objective of GraphTOP to ensure tight relaxation and maintain graph sparsity. Our theoretical analysis indicates that topology-oriented prompting can effectively enhance the pre-trained GNN models for node classification. We conduct comprehensive experiments over five datasets under four pre-training strategies to evaluate the performance of our proposed method. The experimental results validate the superiority of GraphTOP against six baselines.

We summarize the main contributions as follows:

- **Problem formulation.** We formulate and conduct an initial investigation of graph prompting from the perspective of graph topology.

- **Algorithmic design.** We propose GraphTOP, the first topology-oriented prompting framework to adapt pre-trained GNN models by modifying graph topology of the input graph.

- **Theoretical analysis.** We provide theoretical analysis to support that our design of topology-oriented prompting in GraphTOP can effectively enhance pre-trained GNN models for node classification.

- **Experimental evaluation.** We conduct comprehensive experiments on five public datasets under four pre-training strategies. Experimental results validate the effectiveness of our proposed GraphTOP against six baselines for node classification.

## 2 Related works

### 2.1 Graph pre-training

Graph pre-training aims to train powerful GNN models over unlabeled graph data in a self-supervised fashion [44]. Generally, graph pre-training methods can be roughly categorized into two types: contrast-based methods and generation-based methods. Contrast-based methods [34, 39, 45, 49, 54] are developed based on the concept of mutual information (MI) maximization, where the estimated MI between different views of similar objects is maximized. For example, DGI [39] maximizes MI between the local and global instance views. SimGRACE [45] constructs contrastive views from

the perturbed version of GNN models. Generation-based methods [12, 13, 14, 19, 27, 33, 35] focus on reconstructing specific information from graph data, such as graph structure and node features. For instance, GraphMAE [12] pre-trains GNN models by reconstructing masked node features. Meanwhile, previous graph prompting studies [27, 35] also adopt link prediction as the pre-training strategy.

## 2.2 Graph prompting

Graph prompting adapts pre-trained GNN models to close the objective gap between pre-training and downstream tasks. The intuition is to modify the input graph with learnable prompt vectors for downstream tasks without tuning the pre-trained GNN models. For example, GPPT [35] pre-trains a GNN model via link prediction and adapts it for node classification as the downstream task. It bridges the objective gap between link prediction and node classification by converting node classification to link prediction. GraphPrompt [27] and its variant GraphPrompt+ [50] design prompt vectors as feature weights and apply them to node (hidden) representations. GPF-plus [5] mainly focuses on graph classification as the downstream task and learns to manipulate the input graph by adding extra learnable prompt vectors to node features. All-in-one [36] unifies various downstream tasks as graph-level tasks and similarly designs prompt vectors to modify node features. MultiGPrompt [52] instead inserts prompt vectors into node representations at each hidden layer. ProNoG [51] investigates prompt design for non-homophilic graphs and learns prompt vectors as feature weights based on subgraph representations. While the above graph prompting methods have demonstrated remarkable performance in adapting pre-trained GNN models, they are largely feature-oriented and consistently overlook the potential of topology-oriented prompting design.

# 3 Preliminaries

## 3.1 Graph neural networks

An attributed graph can be denoted as $\mathcal{G} = (\mathcal{V}, \mathcal{E})$ where $\mathcal{V} = \{v_1, v_2, \cdots, v_n\}$ is the set of $n$ nodes and $\mathcal{E} \subset \mathcal{V} \times \mathcal{V}$ is the edge set. It can also be represented as $\mathcal{G} = (\mathbf{A}, \mathbf{X})$. Here, $\mathbf{A} \in \{0, 1\}^{n \times n}$ denotes the adjacency matrix where $a_{ij} = 1$ iff $(v_i, v_j) \in \mathcal{E}$. $\mathbf{X} \in \mathbb{R}^{n \times d_x}$ denotes the feature matrix where the $i$-th row $\boldsymbol{x}_i \in \mathbb{R}^{d_x}$ is the $d_x$-dimensional feature vector of node $v_i \in \mathcal{V}$. $\mathcal{N}(v_i)$ denotes the set of node $v_i$'s neighboring nodes. GNN models leverage node features and graph structure to learn a $d_h$-dimensional representation vector $\boldsymbol{h}_i \in \mathbb{R}^{d_h}$ for each target node $v_i \in \mathcal{V}$. Generally, GNN models follow the message-passing mechanism, in which the representation of node $v_i$ is iteratively updated by aggregating the representations from its neighboring nodes. More concretely, a GNN model $f$ updates the representation of node $v_i \in \mathcal{V}$ at layer $l$ by

$$\boldsymbol{m}_i^{(l)} = \texttt{AGGREGATE}^{(l)}\left(\left\{\boldsymbol{h}_j^{(l-1)} : v_j \in \mathcal{N}(v_i)\right\}\right), \ \boldsymbol{h}_i^{(l)} = \texttt{COMBINE}^{(l)}\left(\boldsymbol{h}_i^{(l-1)}, \boldsymbol{m}_i^{(l)}\right), \quad (1)$$

where $\boldsymbol{h}_i^{(l)} \in \mathbb{R}^{d_l}$ denotes the $d_l$-dimensional representation of node $v_i$ at layer $l$, and we initialize $\boldsymbol{h}_i^{(0)} = \boldsymbol{x}_i$. $\texttt{AGGREGATE}^{(l)}(\cdot)$ represents the aggregation operation extracting the neighboring information of node $v_i$, and $\texttt{COMBINE}^{(l)}(\cdot)$ represents the combination operation integrating the previous representation of node $v_i$ and its neighboring information at layer $l$. The ultimate representation $\boldsymbol{h}_i$ of node $v_i$ after the final layer of the GNN model can be used for diverse downstream tasks. In this study, we focus on node-level downstream tasks, particularly on node classification, i.e., predicting the class label $y_i$ of each target node $v_i$.

## 3.2 Graph prompting

In this study, we mainly center on graph prompting for node-level downstream tasks, e.g., node classification. Given a GNN model pre-trained by a pre-training task, graph prompting aims to adapt the pre-trained GNN model for node classification by learning a graph transformation with trainable prompts to modify the input graph without tuning the pre-trained GNN model. Let $f_{\theta^*}$ denote the pre-trained GNN model with parameters $\theta^*$. Given the input graph $\mathcal{G}$ with a labeled node list $\mathcal{V}_L \subset \mathcal{V}$, the graph transformation $t_\phi$ transforms it to a prompted graph $\tilde{\mathcal{G}} = t_\phi(\mathcal{G})$ where $\phi$ contains learnable prompt parameters. By tuning the prompt parameters in $\phi$, the pre-trained GNN model $f_{\theta^*}$ can generate suitable node representations for node classification through a trainable

classifier $g$ parameterized by $\omega$. Mathematically, we can train $\phi$ and $\omega$ by optimizing the empirical loss minimization problem of graph prompting defined as

$$\min_{\phi,\omega} \mathcal{L}_P(\phi, \omega) = \frac{1}{|\mathcal{V}_L|} \sum_{v_i \in \mathcal{V}_L} \ell\left(g_\omega\left(\left[f_{\theta^*}\left(\tilde{\mathcal{G}}\right)\right]_i\right), y_i\right), \tag{2}$$

where $\ell(\cdot, \cdot)$ denotes the cross-entropy loss for node classification. The main goal of graph prompting is to figure out the optimal graph transformation $t_\phi$ for downstream tasks.

## 4 Methodology

In this section, we present GraphTOP — a graph topology-oriented prompting framework that adapts pre-trained GNN models for node classification by learning to modify graph topology while keeping the pre-trained GNN models frozen. We begin by formulating topology-oriented prompting as an edge rewiring problem. Then, we relax this problem into the continuous probability space via reparameterization and optimize the probabilities through a shared trainable projector. To reduce complexity, we restrict the edge rewiring problem to multi-hop local subgraphs. Finally, we present the complete optimization objective in GraphTOP, designed to ensure tight relaxation and preserve graph sparsity.

### 4.1 Topology-oriented prompting

Graph prompting learns to transform the input graph $\mathcal{G} = (\mathbf{A}, \mathbf{X})$ to a prompted one $\tilde{\mathcal{G}}$. Ideally, the prompted graph $\tilde{\mathcal{G}}$ can generate informative node representations through the pre-trained GNN model $f_{\theta^*}$ and is more suitable for downstream tasks, such as node classification. Unlike previous feature-oriented graph prompting studies [5, 36] that aim to transform the input graph with learnable prompt vectors applied to node features (i.e., $\mathbf{X}$), topology-oriented prompting designs learnable prompts to manipulate graph topology (i.e., $\mathbf{A}$). As a result, we will obtain the prompted graph $\tilde{\mathcal{G}} = (\mathbf{S}, \mathbf{X})$ with the prompted graph topology $\mathbf{S} \in \{0,1\}^{n \times n}$.

To achieve this, we formulate topology-oriented prompting as an edge rewiring problem. Specifically, given each pair of nodes $v_i \in \mathcal{V}$ and $v_j \in \mathcal{V}$ $(v_i \neq v_j)$, the goal is to obtain a learnable binary edge selector $s_{ij} \in \mathbf{S}$ that determines whether an edge exists between them. In other words, edge $(v_i, v_j)$ belongs to the prompted edge set iff $s_{ij} = 1$. Mathematically, we can reformulate the problem of graph prompting in Equation (2) as

$$\min_{\mathbf{S},\omega} \mathcal{L}_P(\mathbf{S}, \omega) = \frac{1}{|\mathcal{V}_L|} \sum_{v_i \in \mathcal{V}_L} \ell\left(g_\omega\left([f_{\theta^*}(\mathbf{S}, \mathbf{X})]_i\right), y_i\right), \text{ s.t. } \mathbf{S} \in \{0,1\}^{n \times n}. \tag{3}$$

### 4.2 Prompt reparameterization

Solving the problem in Equation (3) is challenging since the loss function is discrete with respect to binary edge selectors in $\mathbf{S}$, making it intractable and difficult to apply in practice. In this study, we propose to address this issue through edge rewiring reparameterization. When we treat the edge selectors in $\mathbf{S}$ as binary random variables and reparameterize the problem in Equation (3) with respect to their distributions, it can be relaxed into an expected loss minimization problem over the classifier weight and probability spaces, which is continuous. More concretely, we treat each edge selector $s_{ij} \in \mathbf{S}$ as a Bernoulli random variable following the Bernoulli distribution with probability $p_{ij}$. During the forward pass, each edge selector $s_{ij}$ is sampled from the Bernoulli distribution with probability $p_{ij}$ to be 1 and $1 - p_{ij}$ to be 0. Let $\mathbf{P}$ represent the probability matrix. The problem of topology-oriented prompting in Equation (3) can be relaxed into the expected loss minimization problem as

$$\min_{\mathbf{P},\omega} \mathbb{E}_{\mathbf{S} \sim \text{Bern}(\mathbf{P})} \left[\mathcal{L}_P(\mathbf{S}, \omega)\right], \text{ s.t. } \mathbf{P} \in [0,1]^{n \times n}. \tag{4}$$

Therefore, we reformulate the edge rewiring problem as a continuous problem in Equation (4).

However, solving the above problem via gradient descent remains challenging. The difficulty lies in computing the gradient of the expected loss with respect to the probability matrix $\mathbf{P}$, as Bernoulli sampling is non-differentiable. To make the probability matrix trainable, we propose to reparameterize the Bernoulli sampling in graph prompting via Gumbel-Softmax reparameterization [17, 30, 59, 60]. Before introducing this, we first present the following theorem.

**Theorem 1.** *Given two random variables $G_1$ and $G_2$ that follow the Gumbel distribution* $\text{Gumbel}(0,1)$*, for any probability $p_{ij} \in \mathbf{P}$, we have*

$$\text{Pr}\left(G_1 - G_2 + \log\left(\frac{p_{ij}}{1 - p_{ij}}\right) \geq 0\right) = p_{ij}. \tag{5}$$

The proof of Theorem 1 is provided in Appendix A. Recall that the probability that each edge selector $s_{ij}$ is equal to 1 is also $p_{ij}$. Therefore, we can rewrite the expected loss function in Equation (4) as

$$\mathbb{E}_{\mathbf{G}_1, \mathbf{G}_2}\left[\mathcal{L}_P\left(\mathbb{1}\left(\mathbf{G}_1 - \mathbf{G}_2 + \log\left(\frac{\mathbf{P}}{\mathbf{1}_{n \times n} - \mathbf{P}}\right) \geq 0\right), \omega\right)\right], \tag{6}$$

where $\mathbb{1}(\cdot)$ is the indicator function, and $\mathbf{1}_{n \times n}$ is an $n$-by-$n$ all-ones matrix. $\mathbf{G}_1 \in \mathbb{R}^{n \times n}$ and $\mathbf{G}_2 \in \mathbb{R}^{n \times n}$ are two random matrices where each entry follows the Gumbel distribution $\text{Gumbel}(0,1)$.

Nevertheless, the rewritten expected loss is discrete due to the indicator function. A common solution to this issue is to approximate the expected loss by replacing the indicator function with a sigmoid function. Specifically, the expected loss can be approximated as

$$\mathbb{E}_{\mathbf{G}_1, \mathbf{G}_2}\left[\mathcal{L}_P\left(\sigma\left(\frac{\mathbf{G}_1 - \mathbf{G}_2 + \log\left(\frac{\mathbf{P}}{\mathbf{1}_{n \times n} - \mathbf{P}}\right)}{\tau}\right), \omega\right)\right], \tag{7}$$

where $\sigma(\cdot)$ is the element-wise sigmoid function, and $\tau$ is the temperature annealing parameter that decreases linearly, facilitating the transition from probabilistic approximations to near-deterministic outputs during training.

The final task is the computation of $\mathbf{P}$. Instead of learning each individual probability in $\mathbf{P}$ as free parameters — which is usually hard to train, we propose to obtain the probabilities through a learnable projector shared by nodes. As the GNN model has been well pre-trained, we consider using the informative node representations from the pre-trained GNN model as the input of the projector. Given the representation matrix $\mathbf{H} \in \mathbb{R}^{n \times d_h}$ where the $i$-th row $\boldsymbol{h}_i$ is the representation vector of node $v_i$, we obtain $\mathbf{P}$ through a learnable projector $m$ based on $\mathbf{H}$, i.e., $\mathbf{P} = m(\mathbf{H})$. Specifically, given two nodes $v_i$ and $v_j$ with their representations $\boldsymbol{h}_i$ and $\boldsymbol{h}_j$, the projector computes the probability $p_{ij}$ by

$$p_{ij} = \sigma\left(\mathbf{W}_2\left(\text{ReLU}\left(\mathbf{W}_1\left(\boldsymbol{h}_i + \boldsymbol{h}_j\right)\right)\right)\right), \tag{8}$$

where $\mathbf{W}_1$ and $\mathbf{W}_2$ are trainable weights in the projector $m$. Note that the projector will generate $p_{ij} = p_{ji}$, which is consistent with undirected graphs. The integration strategy of $\boldsymbol{h}_i$ and $\boldsymbol{h}_j$ in $m$ can be altered (e.g., concatenation) to handle directed graphs. Therefore, the final problem of topology-oriented prompting can be formulated as

$$\min_{\phi, \omega} \mathcal{L}_P(\phi, \omega) = \frac{1}{|\mathcal{V}_L|} \sum_{v_i \in \mathcal{V}_L} \ell\left(g_\omega\left([f_{\theta^*}(\mathbf{S}, \mathbf{X})]_i\right), y_i\right), \text{ where } \mathbf{S} = \sigma\left(\frac{\mathbf{g}_1 - \mathbf{g}_2 + \log\left(\frac{m_\phi(\mathbf{H})}{\mathbf{1}_{n \times n} - m_\phi(\mathbf{H})}\right)}{\tau}\right). \tag{9}$$

Here, $\phi = \{\mathbf{W}_1, \mathbf{W}_2\}$ represents the prompt parameters that we aim to learn. $\mathbf{g}_1 \in \mathbb{R}^{n \times n}$ and $\mathbf{g}_2 \in \mathbb{R}^{n \times n}$ are two matrices where each entry is sampled from $\text{Gumbel}(0,1)$ per iteration. Since the representation matrix $\mathbf{H}$ is generated by the pre-trained GNN model and fixed during training, we can compute it before the adaptation phase to avoid additional computational costs.

### 4.3 Subgraph-constrained topology-oriented prompting

Although the above formulation is feasible to solve, it still poses challenges in scalability. Since we need to learn an edge selector for each node pair, the number of learnable edge selectors grows overwhelmingly large as the number of nodes in the input graph increases. In GraphTOP, we propose subgraph-constrained topology-oriented prompting to reduce the complexity. In most GNN architectures, the representation of a target node through a pre-trained GNN model primarily depends on its local subgraph [23, 51]. Motivated by this, we consider restricting learning edge selectors for edge rewiring to a multi-hop local subgraph of each target node. More concretely, we first extract the $\rho$-hop subgraph $\mathcal{G}(v_i) = (\mathbf{A}(v_i), \mathbf{X}(v_i))$ for each target node $v_i$. Here, $\mathbf{X}(v_i)$ includes the feature vectors of node $v_i$ and other nodes within $\rho$ steps of node $v_i$, and $\mathbf{A}(v_i)$ indicates the connection between these nodes extracted from the original adjacency matrix $\mathbf{A}$. $\rho$ is a pre-defined hyperparameter. Typically, we set $\rho = 2$ to balance efficiency and effectiveness. Then, the task is

to perform edge rewiring to obtain the prompted subgraph $\tilde{\mathcal{G}}(v_i) = (\mathbf{S}(v_i), \mathbf{X}(v_i))$ based on $\mathcal{G}(v_i)$. Nonetheless, the computation is still expensive if we consider edge connections between each pair of nodes in $\mathcal{G}(v_i)$. Theoretically, the computational cost is approximately $\mathcal{O}(D^{2\rho})$, where $D$ represents the average node degree.

To further reduce the computational cost of edge rewiring, we propose to rewire edges only between the target node and other nodes in $\mathcal{G}(v_i)$. In this way, we only need to consider how the target node connects to other nodes while keeping other edges in the original subgraph $\mathcal{G}(v_i)$ intact. As a result, the computational cost will significantly decrease, reducing to $\mathcal{O}(D^{\rho})$. More specifically, For each pair of nodes $v_j$ and $v_k$ in the subgraph $\mathcal{G}(v_i)$, we can compute $s_{jk} \in \mathbf{S}(v_i)$ by

$$
s_{jk} = \begin{cases} \sigma \left( \dfrac{g_1 - g_2 + \log\left(\frac{p_{jk}}{1-p_{jk}}\right)}{\tau} \right), & \text{if } v_i \in \{v_j, v_k\}, \\ a_{jk}, & \text{otherwise,} \end{cases}
\tag{10}
$$

where $g_1$ and $g_2$ are two values sampled from $\texttt{Gumbel}(0,1)$. Therefore, the objective function in Equation (9) can be reformulated as

$$
\mathcal{L}_P(\phi, \omega) = \frac{1}{|\mathcal{V}_L|} \sum_{v_i \in \mathcal{V}_L} \ell\left(g_\omega\left([f_{\theta^*}\left(\mathbf{S}(v_i), \mathbf{X}(v_i)\right)]_i\right), y_i\right).
\tag{11}
$$

### 4.4 Prompt optimization

While the problem of topology-oriented prompting becomes solvable through the relaxation in Section 4.2, simply solving the problem in Equation (11) cannot guarantee the tightness of the relaxation. Typically, it is unlikely to obtain every probability $p_{ij}$ converging to 0 or 1 after training. As a result, Bernoulli sampling may cause significant fluctuations in graph topology during inference, resulting in unstable classification performance. Therefore, the goal here is to encourage deterministic probabilities (i.e., either 0 or 1) during training. To achieve this, we introduce an extra entropy-based regularization term $\mathcal{L}_E(\phi)$ in the objective function. The intuition of this regularization term is to penalize high-entropy probabilities via entropy minimization [10]. Mathematically, the regularization term $\mathcal{L}_E(\phi)$ can be written as

$$
\mathcal{L}_E(\phi) = \frac{1}{|\mathcal{V}_L|} \sum_{v_i \in \mathcal{V}_L} \left( \frac{1}{|\mathcal{N}_\rho(v_i)|} \sum_{v_j \in \mathcal{N}_\rho(v_i)} e\left(p_{ij}\right) \right), \text{ where } e\left(p_{ij}\right) = p_{ij}\log p_{ij} + (1-p_{ij})\log(1-p_{ij}).
\tag{12}
$$

Here, $\mathcal{N}_\rho(v_i)$ represents the set of nodes within node $v_i$'s $\rho$-hop subgraph except node $v_i$. By minimizing $\mathcal{L}_E$, each probability $p_{ij}$ will be likely to converge to 0 or 1 after training, leading to a deterministic graph topology for inference. As a result, the relaxation in Equation (4) becomes tight.

Meanwhile, the prompted graph may become overly dense, resulting in an almost fully connected graph topology. Such prompted graphs are often impractical for most applications and incur high computational costs [2, 20, 26]. Therefore, it is important to control how sparse the prompted graph is. To achieve this, we introduce another regularization term $\mathcal{L}_S(\phi)$ to constrain the size of connected edges in the prompted graph. More specifically, the regularization term $\mathcal{L}_S(\phi)$ can be formulated as

$$
\mathcal{L}_S(\phi) = \frac{1}{|\mathcal{V}_L|} \sum_{v_i \in \mathcal{V}_L} \left| \frac{\sum_{v_j \in \mathcal{N}_\rho(v_i)} p_{ij}}{|\mathcal{N}_\rho(v_i)|} - \gamma \right|,
\tag{13}
$$

where $\gamma$ is a hyperparameter to control the size of connected edges for each node. Finally, we can write the overall objective function as

$$
\min_{\phi, \omega} \mathcal{L}_P(\phi, \omega) + \lambda_1 \mathcal{L}_E(\phi) + \lambda_2 \mathcal{L}_S(\phi),
\tag{14}
$$

where $\lambda_1$ and $\lambda_2$ are two hyperparameters to balance different loss terms.

## 5 Analysis of GraphTOP

In this section, we present a comprehensive analysis of our proposed framework. The overall algorithm of GraphTOP can be found in Appendix B.

## 5.1 Complexity analysis

Without loss of generality, we take a $K$-layer GCN model as an example for complexity analysis. When we set $\rho = 2$, the representation of each target node is computed based on its 2-hop local subgraph. When the average node degree is $D$, the expected number of nodes within its 2-hop local subgraph is $\mathcal{O}(D^2)$. Accordingly, the expected number of edges within its 2-hop local subgraph is $\mathcal{O}(D^3)$. We assume each layer $l$ of the GCN model has the same hidden size as the feature matrix, i.e., $d_x = d_h = d_l$ for simplicity. In this case, the time complexity of the GCN model is $\mathcal{O}(KD^2d_l^2 + KD^3d_l)$. The extra cost in GraphTOP comes from the computation of the probabilities when generating the prompted graph. The time complexity for computing the probabilities is $\mathcal{O}(D^2d_l^2 + D^2d_l)$. Therefore, we can conclude that computing the probabilities in GraphTOP does not introduce significant additional computational costs compared with GNN models.

## 5.2 Theoretical analysis

In this subsection, we provide a theoretical analysis of how topology-oriented prompting in our GraphTOP benefits pre-trained GNN models for node classification. Following previous graph learning studies [29, 40], our analysis is similarly based on the contextual stochastic block model (CSBM) [4]. Given a random graph $\mathcal{G}$ generated by the CSBM with two node classes $c_1$ and $c_2$, node $v_i$ has the feature vector $\boldsymbol{x}_i$ following a Gaussian distribution $\boldsymbol{x}_i \sim N(\boldsymbol{\mu}_1, \mathbf{I})$ if it is from class $c_1$; otherwise, $\boldsymbol{x}_i \sim N(\boldsymbol{\mu}_2, \mathbf{I})$. Here, we assume $\boldsymbol{\mu}_1 \neq \boldsymbol{\mu}_2$. The edges in $\mathcal{G}$ are generated following an intra-class probability $p > 0$ and an inter-class probability $q > 0$. In other words, each pair of nodes will be linked through an edge with probability $p$ if they are from the same class; otherwise, the probability is $q$. We denote a random graph generated by the CSBM as $\mathcal{G} \sim \mathtt{CSBM}(\boldsymbol{\mu}_1, \boldsymbol{\mu}_2, p, q)$.

We aim to analyze how our edge rewiring design in topology-oriented prompting improves linear separability under pre-trained GNN models. Here, we consider 2-layer linear GCN models [23] for simplicity. We are particularly interested in the expected Euclidean distance between node representations of the two classes after 2-layer GCN operations. Let $\mathtt{Dist}'$ and $\mathtt{Dist}$ denote the expected Euclidean distances with or without our edge rewiring design, respectively. In GraphTOP, we have the following theorem.

**Theorem 2.** *Given a pre-trained 2-layer linear GCN model $f_{\theta^*}$ and a random graph $\mathcal{G} \sim$ $\mathtt{CSBM}(\boldsymbol{\mu}_1, \boldsymbol{\mu}_2, p, q)$, when $p \neq q$, there always exists edge rewiring in the prompted graph by Graph-TOP that satisfies*

$$\mathtt{Dist}' = \frac{p+q}{|p-q|}\mathtt{Dist} > \mathtt{Dist}. \tag{15}$$

The proof of Theorem 2 can be found in Appendix C. Theorem 2 indicates that the expected distance between the two class centroids can be enlarged effectively when GraphTOP alters how every target node connects to the other nodes within its multi-hop local subgraph. Under this circumstance, the representations of nodes from the two classes are more likely to be correctly classified. Therefore, we conclude that our edge rewiring design in GraphTOP can theoretically enhance the classification performance of pre-trained GNN models.

# 6 Experiments

## 6.1 Experimental setup

**Datasets** We adopt five real-world graph datasets from various domains to evaluate the performance of our framework. These datasets include Cora [47], PubMed [47], Amazon [31], Minesweeper [31], and Flickr [53]. Detailed information about these datasets can be found in Appendix E.1.

**Pre-training strategies** To evaluate the compatibility of our framework with different pre-training strategies, we conduct experiments under four representative pre-training strategies. More specifically, we adopt GraphCL [49] and SimGRACE [45] for contrast-based methods. As for generation-based methods, we follow two previous studies — GPPT [35] and GraphPrompt [27] to pre-train GNN models via link prediction. We term them LP-GPPT and LP-GraphPrompt, respectively. More information about these pre-training strategies can be found in Appendix E.2.

Table 1: Accuracy on 5-shot node classification over five datasets. The best-performing method is **bolded**, and the runner-up is underlined.

| Pre-training strategies | Graph prompting methods | Cora | PubMed | Amazon | Minesweeper | Flickr |
|---|---|---|---|---|---|---|
| GraphCL | Linear Probe | $55.69_{\pm 5.74}$ | $67.30_{\pm 6.26}$ | $23.19_{\pm 7.21}$ | $67.59_{\pm 6.30}$ | $29.31_{\pm 8.91}$ |
| | GPPT | $61.50_{\pm 4.49}$ | $65.75_{\pm 3.99}$ | $\underline{24.27}_{\pm 3.74}$ | $65.44_{\pm 8.97}$ | $24.64_{\pm 3.15}$ |
| | ALL-in-one | $52.33_{\pm 4.55}$ | $65.78_{\pm 8.65}$ | $22.82_{\pm 6.09}$ | $63.82_{\pm 8.63}$ | $21.57_{\pm 4.64}$ |
| | GraphPrompt | $\underline{62.12}_{\pm 3.28}$ | $67.01_{\pm 4.56}$ | $21.71_{\pm 2.93}$ | $61.19_{\pm 3.50}$ | $21.92_{\pm 3.72}$ |
| | GraphPrompt+ | $58.91_{\pm 3.12}$ | $66.26_{\pm 5.75}$ | $23.83_{\pm 2.30}$ | $61.64_{\pm 6.36}$ | $24.43_{\pm 4.62}$ |
| | ProNoG | $60.01_{\pm 7.03}$ | $\underline{68.17}_{\pm 4.82}$ | $23.26_{\pm 2.42}$ | $65.48_{\pm 3.40}$ | $26.17_{\pm 5.18}$ |
| | GraphTOP | $\mathbf{63.44}_{\pm \mathbf{4.21}}$ | $\mathbf{68.28}_{\pm \mathbf{4.15}}$ | $\mathbf{27.43}_{\pm \mathbf{7.02}}$ | $\mathbf{68.25}_{\pm \mathbf{7.14}}$ | $\mathbf{30.93}_{\pm \mathbf{9.07}}$ |
| SimGRACE | Linear Probe | $40.68_{\pm 2.29}$ | $54.59_{\pm 6.02}$ | $24.58_{\pm 4.18}$ | $60.58_{\pm 6.42}$ | $26.78_{\pm 5.29}$ |
| | GPPT | $44.83_{\pm 4.67}$ | $52.25_{\pm 5.91}$ | $24.27_{\pm 3.74}$ | $59.62_{\pm 4.80}$ | $22.11_{\pm 3.56}$ |
| | ALL-in-one | $41.11_{\pm 4.92}$ | $51.45_{\pm 4.73}$ | $22.66_{\pm 3.55}$ | $58.11_{\pm 3.82}$ | $21.50_{\pm 4.49}$ |
| | GraphPrompt | $47.02_{\pm 3.87}$ | $55.74_{\pm 5.80}$ | $21.24_{\pm 2.78}$ | $58.72_{\pm 4.37}$ | $19.72_{\pm 4.54}$ |
| | GraphPrompt+ | $\mathbf{51.26}_{\pm \mathbf{4.90}}$ | $55.93_{\pm 6.98}$ | $\mathbf{25.07}_{\pm \mathbf{1.71}}$ | $60.76_{\pm 6.75}$ | $20.79_{\pm 5.65}$ |
| | ProNoG | $42.44_{\pm 2.97}$ | $\underline{55.11}_{\pm 5.98}$ | $22.53_{\pm 2.65}$ | $\mathbf{63.03}_{\pm \mathbf{2.74}}$ | $25.44_{\pm 4.45}$ |
| | GraphTOP | $\underline{50.57}_{\pm 2.91}$ | $\mathbf{56.64}_{\pm \mathbf{5.42}}$ | $\underline{25.67}_{\pm 3.34}$ | $\underline{61.25}_{\pm 5.08}$ | $\mathbf{27.70}_{\pm \mathbf{5.69}}$ |
| LP-GPPT | Linear Probe | $24.40_{\pm 2.83}$ | $42.26_{\pm 5.09}$ | $25.50_{\pm 4.11}$ | $63.22_{\pm 8.16}$ | $23.76_{\pm 4.30}$ |
| | GPPT | $32.08_{\pm 7.66}$ | $44.85_{\pm 4.73}$ | $\underline{28.90}_{\pm 3.50}$ | $63.44_{\pm 8.28}$ | $22.25_{\pm 4.41}$ |
| | ALL-in-one | $26.67_{\pm 6.24}$ | $41.11_{\pm 4.92}$ | $24.49_{\pm 3.51}$ | $59.97_{\pm 4.67}$ | $18.09_{\pm 4.30}$ |
| | GraphPrompt | $30.14_{\pm 2.01}$ | $44.72_{\pm 6.68}$ | $21.88_{\pm 3.56}$ | $61.73_{\pm 6.35}$ | $19.72_{\pm 1.76}$ |
| | GraphPrompt+ | $33.42_{\pm 2.91}$ | $45.17_{\pm 6.91}$ | $24.34_{\pm 1.72}$ | $61.15_{\pm 3.37}$ | $21.02_{\pm 4.22}$ |
| | ProNoG | $\underline{33.71}_{\pm 4.12}$ | $\underline{46.07}_{\pm 3.62}$ | $21.39_{\pm 1.69}$ | $\underline{66.11}_{\pm 4.20}$ | $\underline{24.08}_{\pm 4.10}$ |
| | GraphTOP | $\mathbf{33.97}_{\pm \mathbf{2.43}}$ | $\mathbf{46.52}_{\pm \mathbf{6.18}}$ | $\mathbf{32.41}_{\pm \mathbf{7.18}}$ | $\mathbf{66.67}_{\pm \mathbf{3.83}}$ | $\mathbf{25.95}_{\pm \mathbf{3.17}}$ |
| LP-GraphPrompt | Linear Probe | $50.15_{\pm 5.88}$ | $66.26_{\pm 5.69}$ | $25.00_{\pm 6.78}$ | $65.90_{\pm 7.36}$ | $23.75_{\pm 3.26}$ |
| | GPPT | $52.13_{\pm 7.15}$ | $63.16_{\pm 8.25}$ | $\underline{25.38}_{\pm 5.78}$ | $62.53_{\pm 8.91}$ | $24.16_{\pm 3.88}$ |
| | ALL-in-one | $49.42_{\pm 2.70}$ | $64.73_{\pm 6.46}$ | $21.37_{\pm 3.65}$ | $58.17_{\pm 4.63}$ | $22.10_{\pm 2.92}$ |
| | GraphPrompt | $52.35_{\pm 4.82}$ | $\mathbf{68.16}_{\pm \mathbf{8.23}}$ | $22.76_{\pm 2.81}$ | $58.01_{\pm 3.26}$ | $21.15_{\pm 1.46}$ |
| | GraphPrompt+ | $52.19_{\pm 5.22}$ | $62.19_{\pm 6.70}$ | $24.44_{\pm 1.81}$ | $61.78_{\pm 3.92}$ | $21.48_{\pm 4.09}$ |
| | ProNoG | $\underline{52.49}_{\pm 5.43}$ | $67.68_{\pm 5.02}$ | $23.79_{\pm 2.04}$ | $59.88_{\pm 8.50}$ | $\underline{24.74}_{\pm 1.10}$ |
| | GraphTOP | $\mathbf{53.44}_{\pm \mathbf{4.72}}$ | $\underline{68.14}_{\pm 5.47}$ | $\mathbf{27.07}_{\pm \mathbf{5.84}}$ | $\mathbf{67.09}_{\pm \mathbf{8.45}}$ | $\mathbf{25.03}_{\pm \mathbf{3.84}}$ |

**Baselines**   We include five state-of-the-art graph prompting methods as the baselines of our experiments, including GPPT [35], All-in-one [36], GraphPrompt [27], GraphPrompt+ [50], and ProNoG [51]. Additionally, we also report the performance of tuning linear probes as the classifier based on node representations from pre-trained GNN models without any graph prompting methods (termed *Linear Probe*). More information on these baselines can be found in Appendix E.3.

**Implementation details**   We use a 2-layer GCN [23] as the GNN model for each graph prompting method. The hidden size is 128. All the experimental results are based on the 5-shot setting. Each method is trained using the Adam optimizer [22] with a learning rate of 0.005. The number of epochs is set to 500 for graph prompting. We set $\gamma = 0.5$ in our experiments. We conduct a grid search for $\lambda_1$ and $\lambda_2$. The reported performance is the average result of five runs with different random seeds.

## 6.2 Effectiveness evaluation

We first evaluate the overall performance of our method and other baselines. Table 1 reports the average accuracies on 5-shot node classification over five datasets under four pre-training strategies. According to the results in the table, we first observe that Linear Probe can achieve competitive performance in some cases, although it only trains a classifier during the adaptation phase. For example, it outperforms several baselines on Flickr across four pre-training strategies. Additionally, we observe that ProNoG is a strong baseline compared to other graph prompting methods. It achieves the runner-up position in some cases, such as on four datasets under LP-GPPT. Finally, it is noteworthy that GraphTOP achieves the best performance in most cases compared to other baselines. More specifically, GraphTOP outperforms other baselines in 17 out of 20 experiments. It consistently achieves the best performance on Amazon and Flickr across all four pre-training strategies. These observations validate the effectiveness of GraphTOP in modifying graph topology to adapt pre-trained GNN models for node classification.

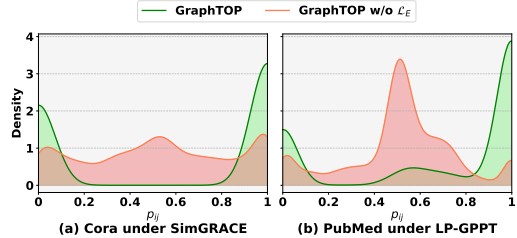
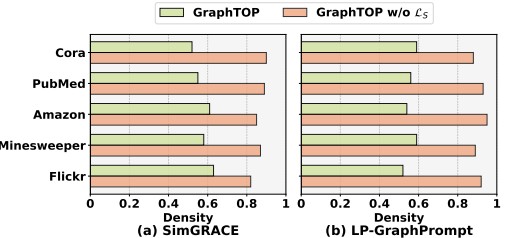

Figure 1: The distribution curves of $p_{ij}$ by Graph-TOP with and without $\mathcal{L}_E$.

Figure 2: The average edge densities of target nodes by GraphTOP with and without $\mathcal{L}_S$.

## 6.3 Analysis of GraphTOP

**Analysis of $\mathcal{L}_E$**  The loss term $\mathcal{L}_E$ ensures tight relaxation of GraphTOP by encouraging each probability to converge to 0 or 1. To evaluate the effectiveness of $\mathcal{L}_E$, we conduct experiments on the probability distribution by GraphTOP and its variant without $\mathcal{L}_E$. Figure 1 illustrates the distribution curves of $p_{ij}$ by GraphTOP with and without $\mathcal{L}_E$. Based on the distribution curves, we observe that the probabilities are not centered around 0 or 1 without $\mathcal{L}_E$. Instead, we may obtain many probabilities around 0.5. In this case, tight relaxation cannot be guaranteed in our reparameterization. However, when we keep $\mathcal{L}_E$ in the objective function, the probabilities are more likely to be close to 0 or 1, which validates the motivation of our design in GraphTOP.

**Analysis of $\mathcal{L}_S$**  In the objective function of GraphTOP, $\mathcal{L}_S$ is designed to control the number of neighboring nodes for each target node by restricting the sparsity of a target node to the threshold $\gamma$ (we set $\gamma = 0.5$ in our experiments). Typically, the edge densities should be forced to $\gamma$ by the loss term $\mathcal{L}_S$. To validate the effectiveness of $\mathcal{L}_S$, we conduct experiments evaluating the edge densities of target nodes when removing $\mathcal{L}_S$. Figure 2 illustrates the average edge densities of target nodes by GraphTOP with and without $\mathcal{L}_S$. From these bar figures, we observe that the average edge densities are consistently very high when we remove $\mathcal{L}_S$ from the objective function. It means that each target node connects to almost all other nodes within its local subgraph, leading to an overly dense graph topology. When we retain $\mathcal{L}_S$ in the objective function, we observe that the average densities by GraphTOP decrease significantly toward 0.5, thereby ensuring the sparsity of the prompted graph. Therefore, we can conclude that the loss term $\mathcal{L}_S$ can effectively avoid overly dense prompted graphs.

**Efficiency analysis**  In Section 4.3, we restrict edge rewiring within the $\rho$-hop local subgraph of each target node. To further reduce the complexity, we propose to rewire edges only between each target node and other nodes within its multi-hop subgraph. To evaluate the efficiency improvement by our design in GraphTOP, we conduct experiments on the running time of GraphTOP and its variant with edge rewiring between each pair of nodes within a multi-hop subgraph (i.e., GraphTOP$_{\text{all\_nodes}}$).

Table 2 shows the running time of the two methods when $\rho = 2$ and $\rho = 3$. From the results in the table, we can observe that GraphTOP$_{\text{all\_nodes}}$ requires much more time compared with GraphTOP. For instance, GraphTOP$_{\text{all\_nodes}}$ needs 651.22 seconds to finish each experiment when $\rho = 2$, which is about $2.97\times$ longer than GraphTOP. Furthermore, when we set $\rho = 3$, GraphTOP$_{\text{all\_nodes}}$ will be out of GPU memory on our server. In contrast, the running time of GraphTOP does not increase significantly. We can observe similar patterns on other datasets. These observations strongly support our design in Graph-TOP by rewiring edges only between each target node with other nodes within its local subgraph.

Table 2: Running time (seconds) of Graph-TOP and its variant over three datasets when $\rho = 2$ and $\rho = 3$ (OOM: out of GPU memory).

| Dataset | $\rho$ | GraphTOP | GraphTOP$_{\text{all\_nodes}}$ |
|---|---|---|---|
| Cora | 2 | 36.52 | 101.70 |
| | 3 | 53.08 | OOM |
| Amazon | 2 | 218.61 | 651.22 |
| | 3 | 252.21 | OOM |
| Minesweeper | 2 | 77.51 | 113.37 |
| | 3 | 78.65 | 199.48 |

**More experimental results**  Due to the page limit, more experimental results, including the influence of $\lambda_1$ and $\lambda_2$, analysis of GPU memory usage, and performance with different numbers of shots, are provided in Appendix F.

# 7 Conclusion

In this study, we conduct a pioneering investigation into graph prompting from the perspective of graph topology. We propose GraphTOP — the first graph topology-oriented prompting framework that adapts pre-trained GNN models by modifying graph topology for downstream tasks, particularly node classification. Extensive experiments over five graph datasets validate the effectiveness of GraphTOP against six baselines under four pre-training strategies.

## Acknowledgments and Disclosure of Funding

This work is supported in part by the National Science Foundation (NSF) under grants IIS-2006844, IIS-2144209, IIS-2223769, IIS-2331315, CNS-2154962, BCS-2228534, and CMMI-2411248, the Office of Naval Research (ONR) under grant N000142412636, the Commonwealth Cyber Initiative (CCI) under grant VV-1Q25-004, and the iPRIME Fellowship Award at UVA.

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

# A  Proof of Theorem 1

**Theorem 1.** *Given two random variables $G_1$ and $G_2$ that follow the Gumbel distribution* $\texttt{Gumbel}(0, 1)$, *for any probability $p_{ij} \in \mathbf{P}$, we have*

$$\Pr\left(G_1 - G_2 + \log\left(\frac{p_{ij}}{1 - p_{ij}}\right) \geq 0\right) = p_{ij}. \tag{16}$$

*Proof.* According to the definition of the Gumbel distribution, the probability density function of $\texttt{Gumbel}(\mu, 1)$ is

$$f(x; \mu) = e^{-(x-\mu) - e^{-(x-\mu)}}. \tag{17}$$

The cumulative distribution function of $\texttt{Gumbel}(\mu, 1)$ is

$$F(x; \mu) = e^{-e^{-(x-\mu)}}. \tag{18}$$

Obviously, we have

$$\Pr\left(G_1 - G_2 + \log\left(\frac{p_{ij}}{1 - p_{ij}}\right) \geq 0\right) = \Pr\left((\log(p_{ij}) + G_1) - (\log(1 - p_{ij}) + G_2) \geq 0\right). \tag{19}$$

Let $x_1 = \log(p_{ij}) + G_1$, $x_2 = \log(1 - p_{ij}) + G_2$. We know

$$x_1 \sim \texttt{Gumbel}(\log(p_{ij}), 1), \ x_2 \sim \texttt{Gumbel}(\log(1 - p_{ij}), 1) \tag{20}$$

Then, we have

$$\begin{aligned}
&\Pr\left((\log(p_{ij}) + G_1) - (\log(1 - p_{ij}) + G_2) \geq 0\right) \\
&= \Pr(x_1 - x_2 \geq 0) \\
&= \int_{-\infty}^{+\infty} \int_{-\infty}^{x_1} f(x_2; \log(1 - p_{ij})) f(x_1; \log(p_{ij})) dx_2 dx_1 \\
&= \int_{-\infty}^{+\infty} F(x_1; \log(1 - p_{ij})) f(x_1; \log(p_{ij})) dx_1 \\
&= \int_{-\infty}^{+\infty} e^{-e^{-(x_1 - \log(1 - p_{ij}))}} \cdot e^{-(x_1 - \log(p_{ij})) - e^{-(x_1 - \log(p_{ij}))}} dx_1 \\
&= \int_{-\infty}^{+\infty} e^{-e^{-(x_1 - \log(1 - p_{ij}))} - (x_1 - \log(p_{ij})) - e^{-(x_1 - \log(p_{ij}))}} dx_1 \\
&= \int_{-\infty}^{+\infty} e^{-(x_1 - \log(p_{ij})) - e^{-x_1} \cdot (e^{\log(1 - p_{ij})} + e^{\log(p_{ij})})} dx_1 \\
&= \int_{-\infty}^{+\infty} e^{-(x_1 - \log(p_{ij})) - e^{-x_1}} dx_1 \\
&= p_{ij} \int_{-\infty}^{+\infty} e^{-x_1 - e^{-x_1}} dx_1 \\
&= p_{ij} \int_{-\infty}^{+\infty} f(x_1; 0) dx_1 \\
&= p_{ij}
\end{aligned} \tag{21}$$

Therefore, we can conclude

$$\Pr\left(G_1 - G_2 + \log\left(\frac{p_{ij}}{1 - p_{ij}}\right) \geq 0\right) = p_{ij}. \tag{22}$$

$\square$

---

**Algorithm 1** GraphTOP

1: **Input:** Pre-trained GNN model $f_{\theta^*}$, initial $g_\omega$ and $t_\phi$, input graph $\mathcal{G}$, training node list $\mathcal{V}_L$, hyperparameters $\lambda_1$ and $\lambda_2$, learning rate $\eta$, training epochs $E$
2: Extract $\rho$-hop local subgraph $\mathcal{G}(v_i)$ for each node $v_i \in \mathcal{V}_L$
3: **for** $e = 1$ **to** $E$ **do**
4:     Anneal temperature by $\tau = \max\left(0.97 \times (1 - e/E) + 0.03, 0.1\right)$
5:     **for** each $v_i$ and its local subgraph $\mathcal{G}(v_i)$ **do**
6:         $\mathbf{S}(v_i) = \mathbf{A}(v_i)$
7:         **for** each node $v_j \in \mathcal{N}_\rho(v_i)$ **do**
8:             Compute $p_{ij} = \sigma\left(\mathbf{W}_2\left(\texttt{ReLU}\left(\mathbf{W}_1\left(\boldsymbol{h}_i + \boldsymbol{h}_j\right)\right)\right)\right)$
9:             Sample $g_1$ and $g_2$ from $\texttt{Gumbel}(0, 1)$
10:             $s_{ij} = \sigma\left(\dfrac{g_1 - g_2 + \log\left(\frac{p_{ij}}{1 - p_{ij}}\right)}{\tau}\right)$
11:             Compute $e\left(p_{ij}\right) = p_{ij}\log p_{ij} + (1 - p_{ij})\log(1 - p_{ij})$
12:         **end for**
13:     **end for**
14:     Compute $\mathcal{L}_P(\phi, \omega) = \frac{1}{|\mathcal{V}_L|}\sum_{v_i \in \mathcal{V}_L}\ell\left(g_\omega\left([f_{\theta^*}\left(\mathbf{S}(v_i), \mathbf{X}(v_i)\right)]_i\right), y_i\right)$
15:     Compute $\mathcal{L}_E(\phi) = \frac{1}{|\mathcal{V}_L|}\sum_{v_i \in \mathcal{V}_L}\left(\frac{1}{|\mathcal{N}_\rho(v_i)|}\sum_{v_j \in \mathcal{N}_\rho(v_i)}e\left(p_{ij}\right)\right)$
16:     Compute $\mathcal{L}_S(\phi) = \frac{1}{|\mathcal{V}_L|}\sum_{v_i \in \mathcal{V}_L}\left|\frac{\sum_{v_j \in \mathcal{N}_\rho(v_i)} p_{ij}}{|\mathcal{N}_\rho(v_i)|} - \gamma\right|$
17:     Update $\omega \leftarrow \omega - \eta\nabla\mathcal{L}_P(\phi, \omega)$
18:     Update $\phi \leftarrow \phi - \eta\nabla\left(\mathcal{L}_P(\phi, \omega) + \lambda_1\mathcal{L}_E(\phi) + \lambda_2\mathcal{L}_S(\phi)\right)$
19: **end for**

---

## B   Overall Algorithm

The overall algorithm of GraphTOP is provided in Algorithm 1.

## C   Proof of Theorem 2

**Theorem 2.** *Given a pre-trained 2-layer linear GCN model $f_{\theta^*}$ and a random graph $\mathcal{G} \sim$ $\texttt{CSBM}(\boldsymbol{\mu}_1, \boldsymbol{\mu}_2, p, q)$, when $p \neq q$, there always exists edge rewiring in the prompted graph by Graph-TOP that satisfies*

$$\texttt{Dist}' = \frac{p + q}{|p - q|}\texttt{Dist}. \tag{23}$$

*Proof.* According to the CSBM, we suppose that the labels of a target node $v_i$'s neighbors will be independently sampled from a neighborhood distribution $\mathcal{D}_{c_1} = \left[\frac{p}{p+q}, \frac{q}{p+q}\right]$ if node $v_i$ is from class $c_1$ or $\mathcal{D}_{c_2} = \left[\frac{q}{p+q}, \frac{p}{p+q}\right]$ if node $v_i$ is from class $c_2$ [29].

Without GraphTOP, the expected feature obtained from the first layer of the GCN operation will be

$$\mathbb{E}_{c_1}[\boldsymbol{h}^{(1)}] = \frac{p}{p+q}\cdot\boldsymbol{\mu}_1 + \frac{q}{p+q}\cdot\boldsymbol{\mu}_2 \tag{24}$$

for nodes from class $c_1$ and

$$\mathbb{E}_{c_2}[\boldsymbol{h}^{(1)}] = \frac{q}{p+q}\cdot\boldsymbol{\mu}_1 + \frac{p}{p+q}\cdot\boldsymbol{\mu}_2 \tag{25}$$

for nodes from class $c_2$. Here, we ignore the linear transformation in the weight matrices of the pre-trained GNN model, as it can be absorbed by the linear classifier.

Similarly, the expected feature obtained from the second layer of the GCN operation will be

$$\mathbb{E}_{c_1}[\boldsymbol{h}^{(2)}] = \frac{p}{p+q}\cdot\mathbb{E}_{c_1}[\boldsymbol{h}^{(1)}] + \frac{q}{p+q}\cdot\mathbb{E}_{c_2}[\boldsymbol{h}^{(1)}] \tag{26}$$

for nodes from class $c_1$ and

$$\mathbb{E}_{c_2}[\boldsymbol{h}^{(2)}] = \frac{q}{p+q} \cdot \mathbb{E}_{c_1}[\boldsymbol{h}^{(1)}] + \frac{p}{p+q} \cdot \mathbb{E}_{c_2}[\boldsymbol{h}^{(1)}] \tag{27}$$

for nodes from class $c_2$. When $p \neq q$, the nodes from the two classes are distinguishable from each other, i.e., $\mathbb{E}_{c_1}[\boldsymbol{h}^{(2)}] \neq \mathbb{E}_{c_2}[\boldsymbol{h}^{(2)}]$.

To evaluate the linear separability of linear classifiers, we calculate the expected distance $\texttt{Dist}$ between the two classes $c_1$ and $c_2$ by

$$
\begin{aligned}
\texttt{Dist} &= \left\| \mathbb{E}_{c_1}[\boldsymbol{h}^{(2)}] - \mathbb{E}_{c_2}[\boldsymbol{h}^{(2)}] \right\| \\
&= \left\| \frac{p-q}{p+q} \cdot \mathbb{E}_{c_1}[\boldsymbol{h}^{(1)}] + \frac{q-p}{p+q} \cdot \mathbb{E}_{c_2}[\boldsymbol{h}^{(1)}] \right\| \\
&= \frac{|p-q|}{p+q} \cdot \left\| \mathbb{E}_{c_1}[\boldsymbol{h}^{(1)}] - \mathbb{E}_{c_2}[\boldsymbol{h}^{(1)}] \right\| \\
&= \frac{|p-q|}{p+q} \cdot \left\| \frac{p-q}{p+q} \cdot \boldsymbol{\mu}_1 + \frac{q-p}{p+q} \cdot \boldsymbol{\mu}_2 \right\| \\
&= \frac{(p-q)^2}{(p+q)^2} \cdot \left\| \boldsymbol{\mu}_1 - \boldsymbol{\mu}_2 \right\|.
\end{aligned}
\tag{28}
$$

Next, we consider the expected distance when rewiring edges with GraphTOP. Since GraphTOP only alters how every target node connects to the nodes within its 2-hop local subgraph, $\mathbb{E}_{c_1}[\boldsymbol{h}^{(1)}]$ and $\mathbb{E}_{c_2}[\boldsymbol{h}^{(1)}]$ are still the expected features for nodes from class $c_1$ and $c_2$, respectively. Let $p'$ and $q'$ denote intra-class and inter-class probabilities of edges between a target node and other nodes after rewiring edges with GraphTOP, respectively. Then, the new expected feature from the second layer of the GCN operation will be

$$\mathbb{E}'_{c_1}[\boldsymbol{h}^{(2)}] = \frac{p'}{p'+q'} \cdot \mathbb{E}_{c_1}[\boldsymbol{h}^{(1)}] + \frac{q'}{p'+q'} \cdot \mathbb{E}_{c_2}[\boldsymbol{h}^{(1)}] \tag{29}$$

for nodes from class $c_1$ and

$$\mathbb{E}'_{c_2}[\boldsymbol{h}^{(2)}] = \frac{q'}{p'+q'} \cdot \mathbb{E}_{c_1}[\boldsymbol{h}^{(1)}] + \frac{p'}{p'+q'} \cdot \mathbb{E}_{c_2}[\boldsymbol{h}^{(1)}] \tag{30}$$

for nodes from class $c_2$. In this case, the new expected distance after rewiring edges with GraphTOP will be

$$
\begin{aligned}
\texttt{Dist}' &= \left\| \mathbb{E}'_{c_1}[\boldsymbol{h}^{(2)}] - \mathbb{E}'_{c_2}[\boldsymbol{h}^{(2)}] \right\| \\
&= \left\| \frac{p'-q'}{p'+q'} \cdot \mathbb{E}_{c_1}[\boldsymbol{h}^{(1)}] + \frac{q'-p'}{p'+q'} \cdot \mathbb{E}_{c_2}[\boldsymbol{h}^{(1)}] \right\| \\
&= \frac{|p'-q'|}{p'+q'} \cdot \left\| \mathbb{E}_{c_1}[\boldsymbol{h}^{(1)}] - \mathbb{E}_{c_2}[\boldsymbol{h}^{(1)}] \right\| \\
&= \frac{|p'-q'|}{p'+q'} \cdot \left\| \frac{p-q}{p+q} \cdot \boldsymbol{\mu}_1 + \frac{q-p}{p+q} \cdot \boldsymbol{\mu}_2 \right\| \\
&= \frac{|p'-q'|}{p'+q'} \cdot \frac{|p-q|}{p+q} \cdot \left\| \boldsymbol{\mu}_1 - \boldsymbol{\mu}_2 \right\|
\end{aligned}
\tag{31}
$$

Then we have

$$\texttt{Dist}' = \frac{|p'-q'|}{p'+q'} \cdot \frac{p+q}{|p-q|} \texttt{Dist}. \tag{32}$$

With the loss term $\mathcal{L}_E$, each probability $p_{ij}$ is forced to be 0 or 1. To ensure that the nodes from the two classes $c_1$ and $c_2$ both have the probability of $p'$ to connect intra-class nodes, we may compute $p_{ij}$ in Equation (8) to be 1 as long as $v_i$ are $v_j$ are from the same class; otherwise, $p_{ij} = 0$. In this case, we will have $p' = 1$ and $q' = 0$ for each target node in the prompted graph. Hence, we can get

$$\texttt{Dist}' = \frac{p+q}{|p-q|} \texttt{Dist}. \tag{33}$$

$\square$

Table 3: Basic information and statistics of graph datasets adopted in our experiments.

| Dataset | #(Nodes) | #(Edges) | #(Features) | Average degree | Homophily ratio | #(Classes) |
|---|---|---|---|---|---|---|
| Cora | 2,708 | 10,556 | 1,433 | 3.90 | 0.810 | 7 |
| PubMed | 19,717 | 88,648 | 500 | 4.49 | 0.802 | 3 |
| Amazon | 24,492 | 93,050 | 300 | 3.80 | 0.380 | 5 |
| Minesweeper | 10,000 | 39,402 | 7 | 3.94 | 0.683 | 2 |
| Flickr | 89,250 | 899,756 | 500 | 10.08 | 0.319 | 7 |

# D   Extension to graph-level tasks

For graph-level tasks, such as graph classification, we can similarly model edge rewiring as Bernoulli random variables. Since graphs in graph-level tasks are typically not very large (usually less than 10K nodes per graph), we do not need to restrict topology-oriented prompting to a multi-hop subgraph but instead apply it to the whole graph. In this case, GraphTOP is simplified for graph-level tasks. We keep empirical evaluation of GraphTOPP on graph-level tasks as our future work.

# E   More details about experimental setup

## E.1   Datasets

We use five real-world graph datasets to evaluate the performance of GraphTOP. The statistics of these datasets can be found in Table 3.

## E.2   Pre-training strategies

We adopt four representative methods for pre-training GNN models in our experiments. These methods are also adopted by the previous graph prompting studies [27, 35, 36]. The details of these pre-training strategies are listed here.

- GraphCL [49] generates two perturbed views of a graph using node dropping and edge perturbation. A GNN model encodes both views into representations, which are then mapped to a latent space using a nonlinear projection head. The contrastive loss is applied to maximize agreement between the two views, optimizing both the GNN model and the projection head.

- SimGRACE [45] constructs a perturbed version of the GNN model by adding Gaussian noise to its parameters. Given an input graph, the perturbed and the original GNN models generate representations that form a positive pair for contrastive learning.

- LP-GPPT [35] randomly masks a subset of edges in the input graph. The model learns to predict whether a given pair of nodes is connected. Negative samples are generated by selecting node pairs that are not linked in the original graph.

- LP-GraphPrompt [27] samples a connected neighbor as one positive node and an unlinked node as one negative node for each target node. The training objective is to maximize the similarity between connected node pairs while minimizing the similarity between unconnected pairs.

We would like to emphasize that designing effective pre-training strategies for training powerful GNNs remains an ongoing challenge in graph learning. Many graph prompting methods [35, 27, 50] adopt link prediction as their pre-training strategies, while one recent study [51] discusses when to use link prediction or contrastive learning for pre-training. Although choosing a better pre-training strategy can yield more powerful GNN models, it is outside the scope of this study: we aim to obtain the best performance on downstream tasks given a pre-trained GNN model, regardless of its pre-training strategy.

## E.3   Baselines

We use six SOTA baselines in our experiments. We provide the details of these baselines as follows.

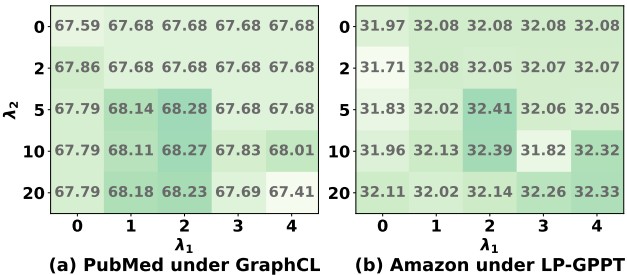

Figure 3: The average accuracies of GraphTOP with different values of $\lambda_1$ and $\lambda_2$.

- Linear Probe only trains a linear classifier during the adaptation phase without any graph prompting design.

- GPPT [35] pre-trains a GNN model via edge masking and link prediction, coupled with a task-specific prompt module that reparameterizes downstream node classification into edge likelihood estimation.

- All-in-one [36] reformulates node/edge tasks as graph tasks via multi-hop subgraphs, using learnable prompt graphs with token vectors, tunable token structures, and feature-similarity-weighted insertion patterns. It aligns downstream tasks with graph-level pre-training via episodic meta-learning over multi-task episodes.

- GraphPrompt [27] bridges the gap between pre-training and downstream tasks through subgraph similarity calculations as a unified template, with a learnable prompt updated during the adaptation phase to incorporate task-specific knowledge for tasks like node and graph classification.

- GraphPrompt+ [50] extends GraphPrompt by introducing prompt vectors within each layer of the pre-trained GNN model, effectively capturing hierarchical information beyond the readout layer to enhance adaptation.

- ProNoG [51]: ProNoG is a graph prompting framework for non-homophilic graphs. It employs a conditional network to generate node-specific prompts based on its multi-hop subgraph.

### E.4 Hardware information

We run our experiments using a server equipped with 512 GB of memory, 128 AMD EPYC 7543 32-core CPUs, and 6 NVIDIA RTX A6000 GPUs, each of which has 48 GB of memory.

## F   More experimental results

### F.1   Influence of $\lambda_1$ and $\lambda_2$

The two hyperparameters $\lambda_1$ and $\lambda_2$ balance different loss terms in the objective function of GraphTOP. To explore their influence on model utility, we conduct grid search for $\lambda_1$ and $\lambda_2$. Other robust optimization strategies, such as Bayesian Optimization and Hyperband [24], can also be used for hyperparameter selection. Figure 3 reports the accuracy results of GraphTOP with different values of $\lambda_1$ and $\lambda_2$ over PubMed under GraphCL and Amazon under LP-GPPT. According to the results, we observe that the performance of GraphTOP varies with different values of both $\lambda_1$ and $\lambda_2$. GraphTOP can obtain the best performance when we set $\lambda_1 = 2$ and $\lambda = 5$ or 10. Additionally, we also notice that the accuracies are not high but quite stable when $\lambda_2 = 0$. We conjecture that the probabilities are mostly driven toward values close to 1 by the loss term $\mathcal{L}_E$, leading to the same densely connected prompted graphs. When $\lambda_2 > 0$, however, the edge densities by GraphTOP are reduced. In this case, detrimental edges will be removed to enhance adaptation for pre-trained GNN models.

Table 4: GPU memory usage (GB) of GraphTOP and its variant when $\rho = 2$ and $\rho = 3$ (OOM: out of GPU memory).

| Dataset | $\rho$ | GraphTOP | GraphTOP$_{\text{all\_nodes}}$ |
|---------|--------|----------|----------------|
| Cora | 2 | 0.44 | 6.41 |
| | 3 | 0.69 | OOM |
| PubMed | 2 | 0.24 | 13.41 |
| | 3 | 2.33 | OOM |
| Amazon | 2 | 0.49 | 6.54 |
| | 3 | 0.58 | OOM |
| Minesweeper | 2 | 0.32 | 0.84 |
| | 3 | 0.36 | 2.79 |

Table 5: Accuracy with different numbers of shots. The best-performing method is **bolded**, and the runner-up is underlined.

| Pre-training Strategies | Graph Prompting Methods | Cora | Minesweeper |
|---|---|---|---|
| **3-shot** | | | |
| | Linear Probe | $51.76_{\pm 2.82}$ | $\underline{63.93_{\pm 6.42}}$ |
| | GPPT | $49.84_{\pm 3.51}$ | $58.77_{\pm 6.21}$ |
| | ALL-in-one | $47.69_{\pm 4.35}$ | $52.40_{\pm 5.58}$ |
| GraphCL | GraphPrompt | $52.76_{\pm 4.94}$ | $55.35_{\pm 8.01}$ |
| | GraphPrompt+ | $50.30_{\pm 4.16}$ | $50.37_{\pm 6.99}$ |
| | ProNoG | $\underline{52.94_{\pm 3.86}}$ | $59.18_{\pm 7.93}$ |
| | GraphTOP | $\mathbf{56.70_{\pm 5.70}}$ | $\mathbf{64.70_{\pm 9.43}}$ |

| Pre-training Strategies | Graph Prompting Methods | Amazon | Minesweeper |
|---|---|---|---|
| **10-shot** | | | |
| | Linear Probe | $\underline{25.36_{\pm 3.89}}$ | $61.60_{\pm 3.93}$ |
| | GPPT | $22.87_{\pm 6.08}$ | $58.90_{\pm 4.70}$ |
| | ALL-in-one | $18.36_{\pm 5.49}$ | $57.64_{\pm 5.74}$ |
| SimGRACE | GraphPrompt | $22.66_{\pm 2.00}$ | $58.65_{\pm 5.34}$ |
| | GraphPrompt+ | $24.06_{\pm 2.47}$ | $61.61_{\pm 5.64}$ |
| | ProNoG | $22.88_{\pm 1.96}$ | $\underline{62.01_{\pm 4.59}}$ |
| | GraphTOP | $\mathbf{26.78_{\pm 4.48}}$ | $\mathbf{63.20_{\pm 8.27}}$ |

| Pre-training Strategies | Graph Prompting Methods | PubMed | Amazon |
|---|---|---|---|
| **20-shot** | | | |
| | Linear Probe | $73.38_{\pm 2.23}$ | $28.50_{\pm 5.70}$ |
| | GPPT | $75.92_{\pm 3.07}$ | $27.54_{\pm 5.38}$ |
| | ALL-in-one | $72.29_{\pm 5.42}$ | $\underline{29.36_{\pm 4.91}}$ |
| LP-GraphPrompt | GraphPrompt | $\underline{76.40_{\pm 1.81}}$ | $25.00_{\pm 1.34}$ |
| | GraphPrompt+ | $70.77_{\pm 2.30}$ | $26.41_{\pm 2.10}$ |
| | ProNoG | $75.55_{\pm 1.69}$ | $26.12_{\pm 1.99}$ |
| | GraphTOP | $\mathbf{77.40_{\pm 5.06}}$ | $\mathbf{32.40_{\pm 4.60}}$ |

## F.2 Results of memory usage

Apart from time efficiency, we are also interested in GPU memory usage by GraphTOP and GraphTOP$_{\text{all\_nodes}}$. Table 4 shows GPU memory usage by GraphTOP and GraphTOP$_{\text{all\_nodes}}$ on four datasets. From the table, we notice that GraphTOP$_{\text{all\_nodes}}$ requires more GPU memory resources compared with GraphTOP. The situation is even more pronounced than what we observed in Table 2. Considering this, our design for rewiring edges only between each target node and other nodes within its multi-hop local subgraph is very essential in GraphTOP.

### F.3  Performance with different numbers of shots

We also conduct experiments with different numbers of shots. Table 5 shows the performance of GraphTOP and other baselines.

## G  Limitations

The theoretical analysis uses the CSBM [4] model to generate random graphs and analyze the linear separability of linear GCN models. Although it follows previous studies in graph learning, the nonlinear separability is also an important issue to explore.

## H  Broader impacts

This study will benefit many real-world applications related to graph prompting, such as anomaly detection and bad actor prediction.

