# OpenReview forum: "GraphTOP: Graph Topology-Oriented Prompting for Graph Neural Networks"
_NeurIPS.cc/2025/Conference — NeurIPS 2025 poster_

### Official Review · Reviewer_6YGZ · 2025-06-29

**Clarity:** 3
**Significance:** 3
**Originality:** 3
**Rating:** 5
**Confidence:** 3

**Summary:**

This paper presents a graph topology-oriented prompting framework, GraphTOP, designed to adapt pre-trained graph neural networks (GNNs) to downstream tasks, with a primary focus on node classification. In contrast to conventional feature-based graph prompting approaches, GraphTOP addresses the topological properties of subgraph rewiring by formulating it as a discrete-to-continuous optimization problem. Specifically, the proposed method transforms discrete topological structures into a continuous probability space amenable to gradient-based training through a carefully designed reparameterization technique. To enhance the framework's robustness, the authors introduce two key components: (1) an entropy regularization term to ensure tight relaxation of the probabilistic formulation, and (2) a sparsity constraint to maintain computational efficiency while preserving meaningful graph structure.The experimental evaluation demonstrates GraphTOP's superiority over six baseline methods across five real-world datasets. Comprehensive ablation studies further validate the effectiveness of the proposed topological prompting mechanism and optimization strategies. The results consistently support the claim that explicitly modeling topological adaptations yields significant performance improvements compared to feature-centric prompting approaches.

**Questions:**

1.How does GraphTOP perform on synthetic datasets? When scaling to larger graphs (exceeding 100,000 nodes), can it maintain a reasonable balance between model effectiveness and computational efficiency?

Scoring impact: Providing experiments on synthetic datasets or large-scale graphs, along with optimization strategies, could potentially improve the rating.

2.The proof in Section 5.2 regarding edge rewiring's effect on inter-class distance expansion relies on linear additivity assumptions. While the authors have noted potential issues under nonlinear conditions with dynamic edge weights, could supplementary theoretical or experimental analysis be provided for nonlinear GCNs (e.g., GAT)?

Scoring impact: Demonstrating that GraphTOP can still enhance GNN training effectiveness under nonlinear representations may lead to rating improvement.

3.The selection of parameters λ₁ and λ₂ shows significant impact on results. Could more robust optimization strategies be proposed?

Scoring impact: The inclusion of more robust optimization approaches may warrant rating consideration.

**Ethical Concerns:**

["NO or VERY MINOR ethics concerns only"]

**Limitations:**

Yes

**Quality:**

3

**Strengths And Weaknesses:**

Strengths

1.The methodological design of this study is rigorous, with meticulous language descriptions demonstrating high quality. For instance, Section 4.2 provides comprehensive theoretical justification for the effectiveness of topology-oriented prompting.

2.Departing from conventional feature-based graph prompting frameworks, this work pioneers topology-oriented graph prompting learning, filling a critical gap in the research field and exhibiting significant originality.

3.The paper features substantial mathematical derivations complemented by clear algorithmic descriptions in the experimental section. Furthermore, the appendices contain complete mathematical proofs of relevant corollaries, substantially enhancing the clarity and rigor of the work.

Weaknesses

1.The experimental evaluation exclusively employs real-world graph datasets, lacking validation on synthetic (artificially constructed) datasets. This omission raises questions about the model's robustness in specialized or edge-case scenarios.

2.The theoretical analysis primarily relies on the CSBM model and linear GCN architectures. The absence of complementary experiments examining generalization to nonlinear scenarios remains an important open question for discussion.

---

> ### Author Rebuttal · Authors · 2025-07-31
>
> We thank the reviewer for providing valuable comments. We hope our point-by-point responses can fully address all your concerns.
>
> ---
> - **W1&Q1**: How does GraphTOP perform on synthetic datasets? When scaling to larger graphs (exceeding 100,000 nodes), can it maintain a reasonable balance between model effectiveness and computational efficiency?
> - **A1**: Thanks for bringing this up. We conduct experiments on a synthetic graph based on the CSBM and provide the results under SimGRACE and LP-GPPT as follows.
>
> |  Graph Prompting Methods  |      SimGRACE     |   LP-GPPT   |
> |---------------------------|----------------------|----------------------|
> |     Linear Probe      |  59.93 $\pm$ 2.76   |  54.07 $\pm$ 3.11   |
> |     GPPT           |  61.39 $\pm$ 4.10   |  53.03 $\pm$ 2.65   |
> |     ALL-in-one       |  58.20 $\pm$ 4.09   |  52.42 $\pm$ 2.46   |
> |      GraphPrompt      |  60.55 $\pm$ 2.05   |  51.91 $\pm$ 3.72   |
> |     GraphPrompt+      |  63.85 $\pm$ 1.64   |  53.05 $\pm$ 1.85   |
> |      ProNoG         |  61.74 $\pm$ 2.49   |  54.60 $\pm$ 2.67   |
> |      GraphTOP        |**65.49 $\pm$ 3.54**  |**57.43 $\pm$ 4.16** |
>
> Additionally, we conduct experiments on ogbn-arxiv with ~170K nodes and provide the results under GraphCL and LP-GraphPrompt as follows.
>
> |  Graph Prompting Methods  |      GraphCL     |   LP-GraphPrompt   |
> |---------------------------|----------------------|----------------------|
> |     Linear Probe      |  21.57 $\pm$ 1.74   |  30.86 $\pm$ 2.63   |
> |     GPPT           |  18.45 $\pm$ 1.83   |  27.51 $\pm$ 1.85   |
> |     ALL-in-one       |  17.85 $\pm$ 3.22   |  16.43 $\pm$ 4.02   |
> |      GraphPrompt      |  21.29 $\pm$ 2.53   |  32.89 $\pm$ 1.84   |
> |     GraphPrompt+      |  21.86 $\pm$ 2.91   |  31.56 $\pm$ 1.24   |
> |      ProNoG         |  20.60 $\pm$ 3.42   |  32.25 $\pm$ 2.57   |
> |      GraphTOP        |**23.54 $\pm$ 2.29**  |**33.86 $\pm$ 2.81** |
>
> ---
> - **W2&Q2**: The proof in Section 5.2 regarding edge rewiring's effect on inter-class distance expansion relies on linear additivity assumptions. While the authors have noted potential issues under nonlinear conditions with dynamic edge weights, could supplementary theoretical or experimental analysis be provided for nonlinear GCNs (e.g., GAT)?
> - **A2**: Thanks for bringing this up. Our theoretical analysis follows classical studies [1, 2] analyzing linear separability under GNNs using linear GCNs because GCNs are the most popular GNNs in graph learning. We will analyze other complex GNNs (e.g., GAT) in our future work.
>
> ---
> - **Q3**: The selection of parameters $\lambda_1$ and $\lambda_2$ shows significant impact on results. Could more robust optimization strategies be proposed?
> - **A3**: Thanks for bringing this up. In our experiments, we conduct grid search on different values of $\lambda_1$ and $\lambda_2$. Their optimal values could vary across datasets and pre-training methods. More robust optimization strategies, such as Bayesian Optimization and Hyperband [3] , could be used for hyperparameter selection.
>
> ---
>
> [1] Ma, Yao, et al. Is Homophily a Necessity for Graph Neural Networks? ICLR 2022.
> [2] Baranwal, Aseem, Kimon Fountoulakis, and Aukosh Jagannath. Graph Convolution for Semi-Supervised Classification: Improved Linear
> Separability and Out-of-Distribution Generalization. ICML 2021.
> [3] Li, Liam, et al. Hyperband: A Novel Bandit-Based Approach to Hyperparameter Optimization. JMLR 2018.

---

> > ### Comment · Reviewer_6YGZ · 2025-08-05
> >
> > Regarding my first question: Thank you for supplementing the experimental results on synthetic and large-scale datasets. Based on the additional data you provided, your proposed model does demonstrate superior average performance compared to some existing architectures on these datasets. I recommend further expanding this line of experimentation in future research (including validation on more diverse datasets) to ensure your model achieves even better generalization capability.
> > However, for my second and third questions, I would expect appropriate modifications and limitations to be explicitly stated in the paper.

---

> > > ### Author Response · Authors · 2025-08-06
> > >
> > > Dear Reviewer 6YGZ,
> > >
> > > Thanks for recognizing our new experimental results. As for your second question, we have already included it in the limitations (see Appendix G). As for your third question, we will include the above discussion in the revised version of our paper.
> > >
> > > Best,
> > > Submission9678 Authors

---

### Official Review · Reviewer_JEMG · 2025-07-01

**Clarity:** 3
**Significance:** 2
**Originality:** 2
**Rating:** 3
**Confidence:** 2

**Summary:**

This paper proposes GraphTOP, the first graph topology-oriented prompting framework that adapts pre-trained Graph Neural Networks by learning to modify the graph’s adjacency structure rather than node features. Through edge rewiring formulated as a differentiable probability optimization problem, GraphTOP significantly improves few-shot node classification performance across multiple datasets and pre-training strategies.

**Questions:**

1. Given that in many graphs (e.g., chemical molecules, transportation networks) the edge structure is determined by strict domain knowledge, how does the method ensure that modifying the adjacency matrix does not introduce unreasonable or invalid pseudo-structures?
2. How does this work compare in practice and motivation to Graph Structure Learning methods (such as SLAPS, GSL, IDGL), which also learn or adapt the graph structure
3. In the experiments with varying numbers of shots, why were the pre-training strategies not kept consistent across comparisons?

**Ethical Concerns:**

["NO or VERY MINOR ethics concerns only"]

**Final Justification:**

After reading the rebuttal and other reviews, I would like to maintain my score.

**Limitations:**

yes

**Quality:**

3

**Strengths And Weaknesses:**

Pros:
1. The design of the method corresponds to the motivation.
2. The time cost of the method is taken into consideration, and constraints are imposed on the method to increase the practicality of the method.
3. Extensive experiments supported the method.


Cons:
1. Assuming that the topological challenge is reasonable enough, in many graphs (such as chemical molecules and transportation networks), the existence of edges is determined a priori, and modifying the adjacency matrix may introduce unreasonable pseudo-structures.
2. The experimental tasks are single and all focus on node classification.
3. Some similar methods may be considered as baselines to compare, or for moderate discussion to distinguish them, such as Graph Structure Learning methods.
4. Regarding the experiments with different numbers of shots, some comparisons are missing, and the experiments in the appendix did not control the pre-training method to remain consistent.

---

> ### Author Rebuttal · Authors · 2025-07-31
>
> We thank the reviewer for providing constructive comments. We hope our point-by-point responses can fully address all your concerns.
>
> ---
> - **C1&Q1**: Assuming that the topological challenge is reasonable enough, in many graphs (such as chemical molecules and transportation networks), the existence of edges is determined a priori, and modifying the adjacency matrix may introduce unreasonable pseudo-structures.
> - **R1**: Thanks for pointing out this insightful question. We agree that topology-oriented prompting will learn the modified adjacency matrix involving pseudo-structures. However, we would like to emphasize that the goal of topology-oriented prompting is to enhance model utility of a pre-trained GNN model for downstream tasks. As a result, the modified adjacency matrix will be more suitable for the pre-trained GNN model, although it may be different from the original one. Similarly, the modified node features by feature-oriented prompting may also include pseudo-features (e.g., the age feature is modified to a negative value). We thank the reviewer for bringing up how to understand the modified data by prompting methods, which is very intriguing but has not been well investigated yet. We think it is also an open question in other fields, i.e., the meaning of prompt tokens in NLP [1] and the meaning of prompt patches in CV [2].
>
> ---
> - **C2**: The experimental tasks are single and all focus on node classification.
> - **R2**: Thanks for pointing it out. We would like to clarify that this paper mainly focuses on node classification as the downstream task. As indicated in Appendix D, the proposed method can be extended to graph-level tasks, which are kept as our future work.
>
> ---
> - **C3&Q2**: Some similar methods may be considered as baselines to compare, or for moderate discussion to distinguish them, such as Graph Structure Learning methods.
> - **R3**: Thanks for bringing this up. We agree that graph structure learning also learns to modify graph structures. However, graph structure learning follows an end-to-end manner by training GNN models via supervised learning. This assumption is fundamentally different from graph prompting where GNN models are pre-trained via unsupervised learning and kept frozen during adaptation. Therefore, we argue that graph structure learning methods are not appropriate baselines in the experiments.
>
> ---
> - **C4&Q3**: Regarding the experiments with different numbers of shots, some comparisons are missing, and the experiments in the appendix did not control the pre-training method to remain consistent.
> - **R4**: Thanks for bringing this up. We would like to clarify that the experiments with different numbers of shots do not aim to compare the performance of one graph prompting method with different numbers of shots. Instead, we are comparing the performance of different graph prompting methods in the same condition (the same pre-trained GNN model and the same number of shots). We use different pre-training strategies here because we hope to provide results in more conditions.
>
> ---
>
> [1] Zhou, Kaiyang, et al. Learning to Prompt for Vision-Language Models. IJCV 2022.
> [2] Jia, Menglin, et al. Visual Prompt Tuning. ECCV 2022.

---

> > ### Comment · Reviewer_JEMG · 2025-08-05
> >
> > Thanks to the authors' detailed response. I still have concern that the topological modifications may violate the original domain priors, so I will maintain my current score.

---

> > > ### Author Response · Authors · 2025-08-07
> > > **A Kind Reminder**
> > >
> > > Dear Reviewer JEMG,
> > >
> > > Thank you so much for reviewing our paper and participating in the discussion period. We have provided our clarification on your concern. We think there might be a misunderstanding on how we understand prompt tuning.
> > >
> > > As the discussion period is ending within two days, we are eager to know whether it has addressed your concern. We look forward to your feedback and are more than happy to answer any further questions.
> > >
> > > Best,
> > > Submission9678 Authors

---

> ### Author Response · Authors · 2025-08-05
>
> Dear Reviewer JEMG,
>
> Thanks a lot for your reply. We would like to respectfully argue that there might be a misunderstanding in your concern about potential violations of domain priors by graph prompting. We provide the following explanation to clarify this.
>
> We first would like to make a clear statement that **mainstream prompting methods are indeed designed for learning to modify the input data that does not exist in practice, even though the modified data may violate domain priors.**
>
> Let's **consider representative graph prompting methods**, e.g., All-in-one [1] and GPF [2]. When we apply them to graph data (e.g., molecular graphs), they will yield atoms and molecules that do not exist in the real world.
> For example, All-in-one [1] generates prompted graphs by learning extra prompt nodes added to the original graphs.
> Typically, each prompt node cannot be mapped to a specific atom. Therefore, **the prompted graph by graph prompting methods [1, 2] cannot represent a real chemical molecule (i.e., domain priors).**
>
> **Moreover, we can get the same conclusion in classic prompting methods from NLP [3] and CV [4]**. For example, CoOp [3] learns extra prompt tokens that are concatenated with the original tokens. While the original tokens represent specific words, the prompt tokens are not able be mapped to words in natural language. As a result, **CoOp [3] will generate the modified sentences that are not linguistically reasonable or consistent with domain priors in natural language.**
>
> In a nutshell, we believe that **your concern about violating domain priors should not be viewed as a weakness of our method.**
> We hope our clarification can fully address your concern. We are happy to provide more explanation if you have any further questions.
>
> Best,
> Submission9678 Authors
>
> [1] Sun, Xiangguo, et al. All in one: Multi-task prompting for graph neural networks. KDD 2023.
> [2] Fang, Taoran, et al. Universal Prompt Tuning for Graph Neural Networks. NeurIPS 2023.
> [3] Zhou, Kaiyang, et al. Learning to Prompt for Vision-Language Models. IJCV 2022.
> [4] Jia, Menglin, et al. Visual Prompt Tuning. ECCV 2022.

---

> ### Author Response · Authors · 2025-08-08
> **Your evaluation is important to us**
>
> Dear Reviewer JEMG,
>
> As the discussion period is ending within 19 hours, we are eager to know whether your concern has been addressed. We think there might be a misunderstanding on how we understand prompt tuning. We look forward to your feedback and are more than happy to answer any further questions.
>
> Best,
> Submission9678 Authors

---

### Official Review · Reviewer_pf7m · 2025-07-02

**Clarity:** 1
**Significance:** 3
**Originality:** 3
**Rating:** 4
**Confidence:** 3

**Summary:**

This work focus on the graph prompt tuning topic, it points out existing methods primarily rely on manipulating node features, which often overlook the structural prompting. So the authors propose the GraphTOP prompting framework that optimizes graph topology by learning to rewire edges within local subgraphs using a Gumbel-Softmax relaxation. The method incorporates entropy and sparsity regularization to ensure structural sparsity and interpretability. To support the design, the authors provide theoretical analysis under the Contextual Stochastic Block Model (CSBM), showing that rewiring improves class separability, and conduct experiments across 5 benchmarks demonstrating consistent gains over existing prompt methods on diverse pre-training strategies.

**Questions:**

1. Is this topology-based prompt compatible with existing feature-based prompt?
any chance they could be combined to make further improvement?

2. Is there any visualized example to illustrate a subgraph before/after re-wiring?

**Ethical Concerns:**

["NO or VERY MINOR ethics concerns only"]

**Final Justification:**

I have read all the author responses, but all the responses are like "we will do xxx", I did not really see the updates. Therefore, I keep my previous score.

**Quality:**

2

**Strengths And Weaknesses:**

Strengths:

1. The proposed GraphTOP achieves good performance
2. The analysis under CSBM provides justification for why graph topology optimization improves downstream task performance.
3. Code is provided.

Weakness

1. Missing related work - This work (https://arxiv.org/pdf/2503.00750) is graph prompting from edge perspective and should be somewhat related.
2. Writing could be improved - the current writing is not easy-to-follow, the paper is presented in a dense and notation-heavy manner, adding some illustrative figure could help.

---

> ### Author Rebuttal · Authors · 2025-07-31
>
> We thank the reviewer for providing valuable comments. We hope our point-by-point responses can fully address all your concerns.
>
> ---
> - **W1**: Missing related work - This work (https://arxiv.org/pdf/2503.00750) is graph prompting from edge perspective and should be somewhat related.
> - **R1**: Thanks for bringing our attention to this excellent work. We will include it in the revised version of our paper.
>
> ---
>
> - **W2**: Writing could be improved - the current writing is not easy-to-follow, the paper is presented in a dense and notation-heavy manner, adding some illustrative figure could help.
> - **R2**: Thanks for pointing it out. We will try to add some illustrative figures to make the paper more readable.
>
> ---
>
> - **Q1**: Is this topology-based prompt compatible with existing feature-based prompt? any chance they could be combined to make further improvement?
> - **A1**: Thanks for bringing this up. Yes, they are compatible and could be combined. It is definitely a promising scheme to combine them for further improvement. We will explore this in our future work.
>
> ---
>
> - **Q2**: Is there any visualized example to illustrate a subgraph before/after re-wiring?
> - **A2**: Thanks for pointing it out. Since we are not allowed to provide figures during rebuttal, we will try to perform the case study in the revised version of our paper.

---

### Official Review · Reviewer_dFHj · 2025-07-02

**Clarity:** 3
**Significance:** 2
**Originality:** 3
**Rating:** 4
**Confidence:** 4

**Summary:**

The paper proposes a topology-oriented prompting strategy to adapt a pre-trained GNN to a new task by modifying the input graph topology.

**Questions:**

Please connect your subgraph-constrained topology-oriented prompting with the line of work of Subgraph GNNs, which also represents related work. See for instance Southern et al., 2025 and all works therein.


Southern et al., Balancing Efficiency and Expressiveness: Subgraph GNNs with Walk-Based Centrality

**Ethical Concerns:**

["NO or VERY MINOR ethics concerns only"]

**Final Justification:**

I maintain my positive score, as I believe a higher score requires a more comprehensive evaluation.

**Limitations:**

yes

**Quality:**

2

**Strengths And Weaknesses:**

The paper is well written and easy to follow. The general idea of leveraging graph topology in prompting is intuitively justified and original.


I think the paper has two main weaknesses:
1. It has been shown that a standard GNN (such as that in Eq. 1) cannot solve link prediction tasks (Zhang et al 2021). Therefore, using a standard GNN for ``pre-training, adaptation'' when pre-training is link prediction is not ideal, as it will not be able to perform the pre-training task well. I think you either should not consider link prediction as pre-training task (either remove it from the paper or move it to the appendix clarifying why theoretically is not a good strategy given the model) or change the GNN model, as shown in Bevilacqua et al., 2025.
2. The evaluation section is weak. The considered datasets are small and outdated. I would recommend to include the ogb datasets (arxiv, proteins, products).


Zhang et al., 2021. Labeling trick: A theory of using graph neural networks for multi-node representation learning.

Bevilacqua et al., 2025. Holographic Node Representations: Pre-training Task-Agnostic Node Embeddings

---

> ### Author Rebuttal · Authors · 2025-07-31
>
> We thank the reviewer for providing valuable comments. We hope our point-by-point responses can fully address all your concerns.
>
> ---
> - **W1**: It has been shown that a standard GNN (such as that in Eq. 1) cannot solve link prediction tasks (Zhang et al 2021). Therefore, using a standard GNN for "pre-training, adaptation" when pre-training is link prediction is not ideal, as it will not be able to perform the pre-training task well. I think you either should not consider link prediction as pre-training task (either remove it from the paper or move it to the appendix clarifying why theoretically is not a good strategy given the model) or change the GNN model, as shown in Bevilacqua et al., 2025.
>
> - **R1**: Thanks for bringing this up. We would like to argue that designing effective pre-training strategies for training powerful GNNs is still an ongoing probelm in graph learning. Although choosing a better pre-training strategy can yield more powerful GNN models, it is outside the scope of this study: we are aiming to obtain the best performance on downstream tasks given a pre-trained GNN model no matter how it is pre-trained. While link prediction may not be the best pre-training strategy, many graph prompting methods [1, 2, 3] adopt link prediction as their pre-training strategies. Additionally, one recent study [4] discusses when to use link prediction or contrastive learning for pre-training. We will discuss this in the revised version of our paper.
>
> ---
>
> - **W2**: The evaluation section is weak. The considered datasets are small and outdated. I would recommend to include the ogb datasets (arxiv, proteins, products).
> - **R2**: Thanks for bringing this up. We conduct experiments on ogbn-arxiv and provide the results under GraphCL and LP-GraphPrompt as follows.
>
> |  Graph Prompting Methods  |      GraphCL     |   LP-GraphPrompt   |
> |---------------------------|----------------------|----------------------|
> |     Linear Probe      |  21.57 $\pm$ 1.74   |  30.86 $\pm$ 2.63   |
> |     GPPT           |  18.45 $\pm$ 1.83   |  27.51 $\pm$ 1.85   |
> |     ALL-in-one       |  17.85 $\pm$ 3.22   |  16.43 $\pm$ 4.02   |
> |      GraphPrompt      |  21.29 $\pm$ 2.53   |  32.89 $\pm$ 1.84   |
> |     GraphPrompt+      |  21.86 $\pm$ 2.91   |  31.56 $\pm$ 1.24   |
> |      ProNoG         |  20.60 $\pm$ 3.42   |  32.25 $\pm$ 2.57   |
> |      GraphTOP        |**23.54 $\pm$ 2.29**  |**33.86 $\pm$ 2.81** |
>
>
> ---
>
> - **Q1**: Please connect your subgraph-constrained topology-oriented prompting with the line of work of Subgraph GNNs, which also represents related work. See for instance Southern et al., 2025 and all works therein.
>
> - **A1**: Thanks for bringing this up. We will include related works of subgraph GNNs in the revised version of our paper.
>
> ---
>
> [1] Liu, Zemin, et al. GraphPrompt: Unifying Pre-Training and Downstream Tasks for Graph Neural Networks. WWW 2023.
> [2] Yu, Xingtong, et al. Generalized Graph Prompt: Toward a Unification of Pre-Training and Downstream Tasks on Graphs. TKDE 2024.
> [3] Sun, Mingchen, et al. GPPT: Graph Pre-training and Prompt Tuning to Generalize Graph Neural Networks. KDD 2022.
> [4] Yu, Xingtong, et al. Non-Homophilic Graph Pre-Training and Prompt Learning. KDD 2025.

---

> > ### Comment · Reviewer_dFHj · 2025-08-04
> >
> > Thank you for the response. I will maintain my positive score (score 4). I believe a higher score requires a more comprehensive evaluation, which encompasses the proteins and products dataset I mentioned in my initial review.

---

> > > ### Author Response · Authors · 2025-08-04
> > >
> > > Dear Reviewer dFHj,
> > >
> > > Thanks a lot for your reply. Due to the time limit, we were able to finish experiments only on ogbn-arxiv within one week. We will conduct experiments on large-scale datasets (the proteins and products datasets) in the coming weeks.
> > >
> > > Best,
> > > Submission9678 Authors

---

### Note · Authors · 2025-08-11

Dear Program Chairs, Senior Area Chairs, Area Chairs, and Reviewers,

We first would like to thank you all for organizing/participating the rebuttal period.

Our paper proposes a novel graph prompting method from an innovative perspective of graph topology and presents comprehensive theoretical analysis of the proposed method. Extensive experiments are conducted to demonstrate the effectiveness of our method. **Most reviewers give positive ratings in their initial reviews and appreciate the contributions of our paper, including *novel algorithmic design*, *solid theoretical analysis*, *superior experimental performance*, and *clear paper writing*.** We believe that this study makes a significant contribution to the field and has the potential to inspire future research on topology-oriented approaches to graph prompting methods.

After the reviewer-author discussion, we think **the only remaining divergence is about how to understand the modified graph data by graph prompting methods from Reviewer JEMG**. Reviewer JEMG thinks our proposed method learns to modify graph topology using tunable prompts, resulting in unreasonable graph data that does not exist in the real world. However, **we would like to respectfully point out that this is a fundamental misunderstanding on our method.** We want to emphasize that not only our proposed method but also most representative graph prompting methods (e.g., All-in-one and GPF) modify graph data to the prompted graph data that does not exist in practice. In fact, the primary goal of graph prompting itself is to learn how to modify graph data, even though the modified data may violate domain priors. Therefore, **we argue that this should not be regarded as a weakness of our study.** We provide a more detailed clarification to explain this and hope the further clarification has fully addressed Reviewer JEMG's concern.

In the end, we would like to express our gratefulness again for the valuable comments from our reviewers. We believe these comments are very constructive and make our paper better.

Best,
Submission9678 Authors

---

### Decision · Program_Chairs · 2025-09-17

**Decision:**

Accept (poster)

**Comment:**

This paper proposes a new graph prompting method that works by changing the graph structure instead of node features. The idea is original and fills an important gap in the area. The authors design a complete framework, GraphTOP, that uses edge rewiring in a way that can be optimized during training. The method is supported by theoretical analysis and tested on several datasets with different pre-training strategies. Most reviewers agree that the paper is technically strong and that the proposed idea is interesting and promising. The experimental results are good and show consistent improvements over many baselines.

One reviewer raised concern about modifying the graph structure in cases where the topology is given by domain knowledge. However, the authors explain clearly that the goal of prompting is not to keep domain constraints but to improve performance by adapting to the pre-trained model. I agree with this explanation and believe the concern comes from a misunderstanding. Other reviewers asked for larger and synthetic datasets, and the authors provided additional experiments that helped support their claims. For these reasons, I recommend to accept this paper.